# A single nanobody neutralizes multiple epochally evolving human noroviruses by modulating capsid plasticity

Wilhelm Salmen[1], Liya Hu[1], Marina Bok [2], Natthawan Chaimongkol[3], Khalil Ettayebi [4], Stanislav V. Sosnovtsev[3], Kaundal Soni[1], B. Vijayalakshmi Ayyar [4], Sreejesh Shanker[1], Frederick H. Neill[4], Banumathi Sankaran [5], Robert L. Atmar[4,6], Mary K. Estes [4,6], Kim Y. Green[3], Viviana Parreño[2] & B. V. Venkataram Prasad [1,4] ✉

Acute gastroenteritis caused by human noroviruses (HuNoVs) is a significant global health and economic burden and is without licensed vaccines or antiviral drugs. The GII.4 HuNoV causes most epidemics worldwide. This virus undergoes epochal evolution with periodic emergence of variants with new antigenic profiles and altered specificity for histo-blood group antigens (HBGA), the determinants of cell attachment and susceptibility, hampering the development of immunotherapeutics. Here, we show that a llama-derived nanobody M4 neutralizes multiple GII.4 variants with high potency in human intestinal enteroids. The crystal structure of M4 complexed with the protruding domain of the GII.4 capsid protein VP1 revealed a conserved epitope, away from the HBGA binding site, fully accessible only when VP1 transitions to a "raised" conformation in the capsid. Together with dynamic light scattering and electron microscopy of the GII.4 VLPs, our studies suggest a mechanism in which M4 accesses the epitope by altering the conformational dynamics of the capsid and triggering its disassembly to neutralize GII.4 infection.

Human noroviruses (HuNoVs), members of the genus *Norovirus* in the family *Caliciviridae*, are the leading causative agents of epidemic and sporadic acute viral gastroenteritis worldwide[1,2]. While most immunocompetent patients recover without treatment, norovirus infection can be life-threatening in infants, the elderly, and people with underlying diseases[3]. It is estimated that HuNoVs cause ~684 million illnesses and ~212,000 deaths annually[4–7]. The direct health system and societal costs are estimated to be over $60 billion per year[8]. Despite the substantial societal and economic burdens caused by HuNoVs, no antivirals or norovirus vaccines are available[7].

Noroviruses (NoVs) are nonenveloped, positive-sense single-stranded RNA viruses with a genome consisting of three open reading frames (ORFs). ORF2 and ORF3 encode the major capsid protein VP1 and the minor structural protein VP2, respectively[9]. The amino acid sequence of VP1 is used to classify NoVs into at least ten genogroups (GI-GX), which are further subdivided into 49 genotypes[10]. Among these genogroups, GI, GII, GIV, GVIII, and GIX infect humans, and the viruses in the GII genogroup and genotype 4 (GII.4) are the most predominant. These viruses exhibit preferential accumulations of mutations within VP1 that have indicated the occurrence of genetic drift and selection with each variant descended from chronological

[1]Verna and Marrs McLean Department of Biochemistry and Molecular Pharmacology, Baylor College of Medicine, Houston, TX, USA. [2]Virology Institute and Technology Innovation, IVIT, CONICET-INTA, Hurlingham, Buenos Aires, Argentina. [3]Caliciviruses Section, National Institute of Allergy and Infectious Diseases, National Institutes of Health, Bethesda, MD, USA. [4]Department of Molecular Virology and Microbiology, Baylor College of Medicine, Houston, TX, USA. [5]Berkeley Center for Structural Biology, Molecular Biophysics and Integrated Bioimaging, Lawrence Berkeley Laboratory, Berkeley, CA, USA. [6]Department of Medicine, Baylor College of Medicine, Houston, TX, USA. ✉e-mail: vprasad@bcm.edu

predecessors[11]. Observed genetic findings and changes in epidemiology indicate population immunity drives the epochal evolution of GII.4 norovirus with the periodic emergence of a variant with new antigenic profiles replacing the previous variant as a means of immune evasion[11], similar to H3N2 influenza A virus[12].

Despite the initial obstacles to HuNoV cultivation, there has been remarkable progress in using human intestinal enteroid (HIE) systems for virus replication to study the determinants of infectivity, innate immune responses, and antibody-mediated neutralization[13–15]. However, there are still challenges in these systems to successfully propagate and obtain the virus in sufficient quantities for structural and biochemical studies, which still rely on virus-like particles (VLPs) produced by the co-expression VP1 and VP2[16]. These VLPs are structurally and immunologically similar to authentic virions. While there are some considerable drawbacks to using VLPs, such as the lack of genomic RNA, which may play a role in differentially stabilizing the virus capsid, the use of these VLPs has been invaluable in understanding the structural, immunological, and biological aspects of many strains of HuNoVs[17].

To date, the structures of several caliciviruses have been determined, including feline calicivirus[18], San Miguel sea lion virus[19], murine norovirus (MNV)[20–22], and HuNoV VLPs[23–28] using X-ray crystallography and high-resolution cryo-electron microscopy (cryo-EM). These structural studies have shown that the capsid of calicivirus virions consists of 90 copies of VP1 dimers assembled with a $T = 3$ icosahedral symmetry[28,29]. Each VP1 subunit consists of an internal N-terminal arm (NTA) and two distinct domains, termed shell (S−) and protruding (P-) domain, separated by a flexible hinge[29] (see Supplementary Fig. 1). As first observed in MNV[22,30,31], recent structural studies[21,23–25,28] on HuNoV VLPs have shown that VP1 can exist in two distinct conformations, the "resting" conformation in which the P-domain closely interacts with the S-domain, and the "raised" conformation in which the P-domain is rotated and raised above the S-domain, which is driven either by the removal of stabilizing ions, as in the case of GII.4 VLPs[28], or with increase in pH, as in case of MNV[21]. The P-domain is further divided into P1 and P2 subdomains, with the distal P2 subdomain involved in recognition of cell attachment factors, which is in the case of GII.4 HuNoV are the histo-blood group antigens (HBGAs) that are also the susceptibility factors[32,33] (see Supplementary Fig. 1) The HuNoV VLPs have been useful in the biochemical epitope mapping and structural characterization of the human-derived neutralizing and non-neutralizing monoclonal antibodies (mAbs)[14,34–43].

In addition to the traditional mAbs, llama-derived single-domain antibodies, also known as 'nanobodies', that recognize the HuNoV P-domain have been identified[38,42,44]. Nanobodies have several advantages over traditional antibodies for their development as immunotherapeutic agents. They are smaller in size (~15 kDa), exhibit higher stability over a wide range of temperatures, and are resistant to protease cleavage[45–47]. There has also been substantial work done on the development of nanobodies against several other viral agents, such as hepatitis B virus[48], influenza virus[49], human immunodeficiency virus[50], poliovirus[51], rotavirus[52], and respiratory syncytial virus[53]. We have previously developed a panel of nanobodies against both prototype GI.1 (Norwalk-1968) and the predominant GII.4 (MD2004) VLPs[44]. Among these nanobodies, we chose M4 as it recognized multiple GII HuNoV strains belonging to genotypes 1, 2, 3, 4, 6, and 7 via ELISA[44] and inhibited GII.4 VLP binding to HBGA and saliva, suggesting that it has a strong potential for further development as a therapeutic agent against HuNoVs[44]. However, whether M4 can inhibit virus replication and how it recognizes the GII HuNoV has remained unclear.

Here, using HIEs, we show that M4 inhibits replication of GII.4 HuNoVs very effectively. To understand the mechanism of the M4-mediated neutralization, we determined the crystal structure M4 in complex with GII.4 P-domain. The structure reveals a conserved epitope among GII HuNoVs, which remarkably overlaps with the epitopes of infection- and vaccine-derived human mAbs[34,54]. Modeling of M4 onto GII.4 capsid structure with VP1 in "resting" and "raised" conformations indicates that M4 binds to the raised VP1 conformation. Along with negative-stain EM these observations suggest that M4 uses a novel neutralization mechanism by restricting the conformational plasticity of the capsid to induce stress and mediate the disassembly of virus particles. Our study provides a molecular basis for the further development of nanobody as a therapeutic agent against HuNoVs.

## Results

### M4 neutralized multiple strains of GII.4 HuNoVs

To examine the neutralization potential of M4, which showed binding to multiple GII VLPs in previous studies[44], we infected HIE cultures with 10% stool filtrates containing either GII.3 or different variants of GII.4 HuNoV. M4 effectively neutralized the infection of all the GII.4 HuNoV variants were used in our studies including GII.4 Sydney, GII.4 New Orleans, and GII.4 Den Haag with an $IC_{50}$ of 53 ng/ml, 56 ng/ml, and 379 ng/ml, respectively (Fig. 1). Interestingly, despite binding to GII.3 VLPs[44], M4 did not neutralize GII.3 HuNoV infection.

### Crystal structure shows molecular interactions between M4 and GII.4 P-domain

To define the epitope recognized by M4, we determined the crystal structure of M4 in complex with the GII.4 Sydney P-domain at a resolution of 2.87 Å (Table 1, Fig. 2a). Two molecules of M4 bind symmetrically to the P-domain dimer with each epitope composed of a buried accessible surface of area (ASA) of 673.7 Å$^2$ on subunit A and 224.1 Å$^2$ on subunit B. The M4-binding site, on the side of the P-domain dimer closer to the S-domain in the context of the capsid structure, is away from the HBGA-binding site, indicating neutralization by M4 is not by direct blocking of the HBGA (Fig. 2a). A superposition of the structures of GII.4 Sydney P-domain alone and in complex with M4 and in complex with H-type HBGA showed that M4 binding does not result in any significant conformational changes in the P-domain, with a root mean square deviation of 0.68 Å between matching Cα atoms, and the HBGA-binding site is not affected (Fig. 2b).

The crystal structure of M4 in complex with GII.4 P-domain reveals the molecular details of the interactions. M4 binding near the P-domain dimeric interface involves residues from each of the subunits through a network of hydrogen-bonding and hydrophobic interactions. The majority of these interactions on each side of the dimer involve one of the subunits with residues from both P1 and P2 subdomains (Supplementary Fig. 1) participating in the interactions (Fig. 2c–e). The paratope in M4 consists of eleven residues from all three complementarity-determining regions (CDRs), with residues R99, R100, D101, L102, R105, F106 in CDR3 contributing to most of the interactions with the P-domain (Fig. 2c, d). For instance, R100 in the CDR3 of M4 hydrogen bonds with L272, G274, and T276 in the P-domain subunit, R99 and R105 in the CDR3 hydrogen bond with the side chains D269 in subunit A and E236 in subunit B of the P-domain dimer, respectively (Fig. 2e). Residues D101, L102, and F106 in CDR3 make van der Waals contacts with residues L232, T233, G270, L273, Y462, P480, and Y514 of the P-domain (Fig. 2e). In addition to CDR3, residues I31 in CDR1 and T52 and G54 in CDR3 are in contact with residues K493, H417, and E316 of the P-domain (Fig. 2e). Furthermore, non-CDR residues Q1 and S27 of M4 also interact with A465 and K493 of the P-domain via hydrogen-bonding and hydrophobic interactions (Fig. 2d).

### M4-binding site is highly conserved among GII HuNoVs

The alignment of the P-domain amino acid sequences of 26 GII genotypes (Fig. 3) shows that the identified M4-binding site in our crystallographic structure is highly conserved (72% to 94% identity),

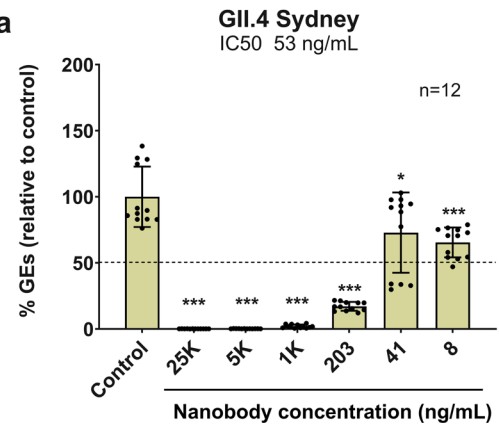

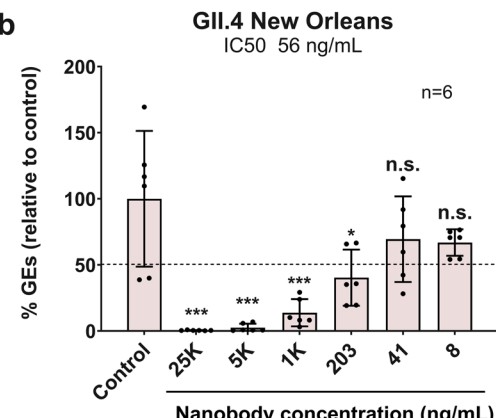

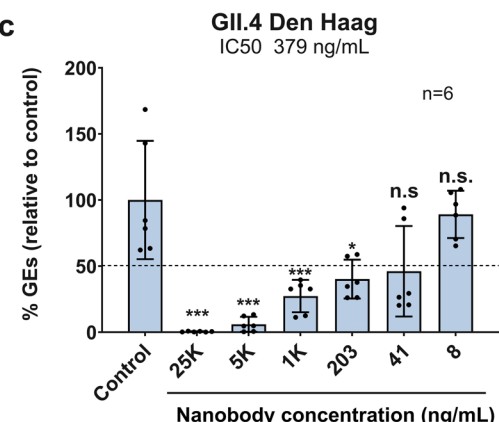

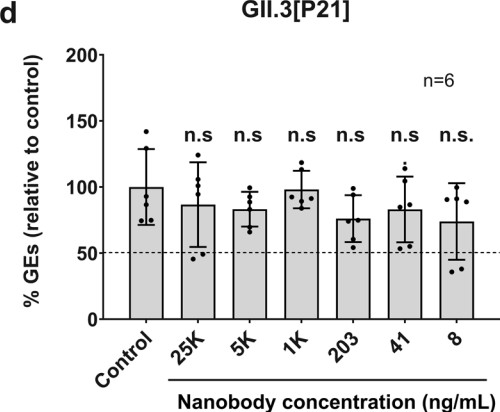

**Fig. 1 | M4 nanobody neutralizes GII.4 variants, but not GII.3, in human intestinal enteroid system.** Serial dilutions of M4, in differentiation medium supplemented with 500 μM GCDCA, were preincubated with equal volume of the same medium containing 100 TCID$_{50}$ of HuNoV for 1 hr at 37 °C. The M4/virus mixtures were inoculated in differentiated J4$^{FUT2}$ HIE monolayers. Viral replication was measured by RT-qPCR after 24 hpi. Percent reduction in viral genome equivalents (GEs) relative to the medium control (100%) was determined. The dotted line represents 50% neutralization. Error bars denote standard deviation and data bars represent the mean of the data collected from the specified number of wells (six wells for **b**–**d**, and 12 replicates for **a**). Significance relative to the control was determined using two-tailed Student's *t* test. Exact *ρ* values from left to right: **a** ***$ρ$ = 3.9E-13, 4.0E-13, 5.9E-13, 1.9E-11, 1.1E-4, *$ρ$ = 0.02; **b** ***$ρ$ = 7.8E-4, 9.2E-4, 2.4E-3, *$ρ$ = 0.02; **c** ***$ρ$ = 2.8E-4, 4.6E-4, 3.3E-3, *$ρ$ = 0.01; **d** all differences are n.s., not significant.

consistent with our previous finding that the M4 binds to multiple GII VLPs[44]. Interestingly, 72% conservation of the epitope in the GII.3 P-domain is on the lower side with five residue changes including V271E, L273M, E316D, H417N, and A465S. The epitope region shows only 44% conservation with the VP1 sequence of GI.1 Norwalk virus, which is consistent with our previous result showing that M4 does not bind to Norwalk VP1.

## M4 binds to the GII.4 VLP in the "raised" conformation

Recent X-ray crystallographic and cryo-EM studies of GII.4 VLP shows that VP1 can exist in "resting" and "raised" conformations[28]. To investigate the binding of M4 to the VP1 dimers in the context of the GII.4 capsid structure, we superimposed the M4-GII.4 P-domain complex onto the GII.4 VLP structure in the "resting" (Fig. 4a–d) and "raised" (Fig 4e–h) states. The structural modeling shows that the M4-epitope is occluded in the "resting" form, and M4 clashes with the neighboring VP1 subunits, suggesting that M4 is unlikely to bind to the P-domain of VP1 in the "resting" state (Fig. 4b). In contrast, in the "raised" T = 3 state, when the P-domain is raised above the shell domain, the epitope becomes accessible for M4 binding (Fig. 4f). The modeling further suggests that the binding of M4 to the "raised" P-domain would prevent it from rotating and descending back onto the S-domain to the more stable "resting" conformation thereby inducing stress on the stability of the capsid leading to particle disassembly (Fig. 4i).

## M4 induces particle disassembly of GII.4 and GII.3 VLPs with different kinetics

To test the hypothesis that M4 binding may affect particle stability leading to disassembly, we performed negative-stain EM, dynamic light scattering (DLS), and sedimentation velocity-analytical ultra-centrifugation (SV-AUC) experiments with GII.4 VLPs treated with and without M4 in PBS. Negative-stain EM data suggest that incubation of M4 with GII.4 VLPs leads to particle disintegration, resulting in smaller structures such as T = 1 particles and capsid fragments (Fig. 4j). Because M4 did not neutralize GII.3 HuNoV in HIEs, we examined whether M4 has a similar effect on GII.3 VLPs using the same experimental procedure. This EM analysis suggested that M4 has a similar effect on GII.3 VLPs (Fig. 4j). Consistent with EM analysis, the DLS data for GII.4 and GII.3 VLPs treated with M4 indicate a significant reduction in the distribution of particle diameters which also likely involves capsid fragments (Supplementary Fig. 2). In the case of GI.1 VLPs, used as a negative control, as the M4 epitope is not conserved in GI.1 (Supplementary Fig. 2C), the diameters remained the same following incubation with M4. To further analyze the capsid disassembly more quantitatively, we performed SV-AUC, of GII.4 and GII.3 VLPs (Supplementary Fig. 3) before and after incubating with M4. When GII.4 and GII.3 VLPs were treated with M4, we observed clear shifts in sedimentation coefficients that indicated a substantial reduction in the molecular weights of the particle fragments. In the case of GII.4 VLPs, we observed a slightly heterogeneous population

**Table 1 | Summary of X-ray crystallography data collection and refinement statistics**

| PDB ID | 8GOW |
| --- | --- |
| Wavelength (Å) | 0.99994 |
| Resolution range (Å) | 47.55–2.87 (2.97–2.87) |
| Space group | P 1 2₁ 1 |
| Unit cell | |
| a, b, c (Å) | 103.5, 90.7, 107.2 |
| α, β, γ (°) | 90, 113.3, 90 |
| Unique reflections | 41401 (4078) |
| Multiplicity | 2.0 (2.0) |
| Completeness (%) | 98.66 (98.26) |
| Mean I/sigma(I) | 5.41/1.95 |
| Wilson B-factor (Å²) | 31.33 |
| R-merge (%) | 9.11 (33.4) |
| R-work (%) | 25.05(33.53) |
| R-free (%) | 29.39(38.28) |
| Number of non-hydrogen atoms | 12,812 |
| Macromolecules | 12810 |
| Ligands | 0 |
| Solvent | 2 |
| Protein residues | 1653 |
| R.M.S. deviations | |
| Bond lengths (Å) | 0.002 |
| Bond angles (°) | 0.56 |
| Ramachandran | |
| Favored (%) | 97.16 |
| Allowed (%) | 2.84 |
| Disallowed (%) | 0 |
| Average B-factor (Å²) | 34.01 |
| Macromolecules | 34.01 |
| Solvent | 29.08 |

Statistics for the highest-resolution shell are shown in parentheses.

of VLPs in the untreated sample that is consistent with molecular weights of capsids with $T = 1$ (black peak I), $T = 3$ (black peak II), and $T = 4$ (black peak III) symmetries (Supplementary Fig. 3A). However, despite showing heterogeneity, when GII.4 VLPs were treated with M4 we observed a significant reduction in percent population of $T = 3$ VLPs (red peak IV), complete loss of $T = 4$ fraction, and the emergence of smaller fractions (red peaks I, II, III), clearly indicative of particle disintegration (Supplementary Fig. 3A). For the case of GII.3 VLPs, we observed complete disappearance of peaks corresponding to molecular weights consistent with $T = 3$ VLPs (black peak IV) and the emergence of peaks with lower molecular weights (red peaks I, II III) (Supplementary Fig. 3B). Similar experiment with GI.1 VLP treated with M4, used as a negative control, did not show any change in sedimentation coefficients (Supplementary Fig. 3C). Together, these experiments clearly demonstrate that M4 triggers disassembly of both GII.4 and GII.3 VLPs.

Further, to examine if there are differences in the kinetics of M4 binding to the GII.4 and GII.3 P-domains, we used biolayer interferometry (BLI). These experiments show that M4 binds to both the GII.4 and GII.3 P-domains with strong binding affinity of $K_D$ of $<1 \times 10^{-12}$ M and $K_D$ of $4.8$–$7.8 \times 10^{-10}$ M, respectively (Supplementary Fig. 4), despite 72% conservation in the M4 epitope. The modest differences in the binding affinities between GII.4 and GII.3 are consistent with the observation that there are only two non-conserved mutations (V271E and A465S) in the M4-binding regions of these

viruses. Despite exhibiting similar binding on rates, the off rates of M4 binding to GII.4 and GII.3 VLPs were significantly different with $K_{dis} = <1 \times 10^{-7}$ per second for GII.4 and $K_{dis} = 1.68$–$1.90 \times 10^{-3}$ per second for GII.3 (Supplementary Fig. 4). This difference suggests that GII.3 VLP disassembly has a slower kinetics compared to that of GII.4 VLPs, and further the kinetics of particle disassembly depends upon the threshold of stably bound M4 required to trigger the disassembly.

## M4 inhibits GII.4-induced endocytosis on HIEs

To test if M4 affects uptake and entry of GII.4 norovirus, we performed endocytosis assays to measure the induction of endocytosis by VLPs in the presence of M4. The effect of M4 nanobody on GII.4-induced endocytosis was confirmed by monitoring uptake of the dye FM1-43FX using epifluorescence microscopy (Supplementary Fig. 5). FM1-43FX is an endocytosis tracking dye that labels extracellular membranes and fluorescent puncta are observed following endocytosis, indicating endocytic trafficking[17]. GII.4 VLP-induced endocytosis (represented by the number of fluorescent puncta) is reduced in the presence of M4 nanobody demonstrating that binding of M4 to GII.4 VLP blocks its entry into the HIEs.

## Discussion

The enormous genetic and antigenic diversity across circulating strains of HuNoV makes it challenging to identify broadly cross-reactive and neutralizing antibodies for the development of immunotherapeutic agents. In the case of GII.4 HuNoV, which accounts for most episodic events worldwide, this is further exacerbated by the periodic emergence of new variants with altered HBGA specificity, allowing these variants to escape herd immunity[11]. In this study, from a panel of llama-derived norovirus-specific nanobodies, we discovered that nanobody M4 neutralizes multiple GII.4 variants with high potency in HIEs potentially using a novel mechanism by targeting a conserved region in the P-domain that is distant from the HBGA-binding site.

To date, there have been several crystal structures of antibodies in complex with the P-domain of GI and GII HuNoV VP1, which show varied antibody recognition patterns. The first pattern is the binding of a human-derived antibody to the P-domain of GI HuNoV at a site that overlaps with the HBGA-binding site indicating that the neutralization mechanism of such antibodies is by directly blocking cell attachment[36]. In the case of GII HuNoVs in general and GII.4 variants in particular, such antibodies that directly block HBGA-binding are unlikely to be broadly cross-reactive, as the epitopes surrounding the HBGA-binding site exhibit sequence variability[55–57]. The second pattern is antibody binding to the sides of the P-domain dimer away from the HBGA-binding site but still blocking cell attachment to neutralize infection by mediating particle aggregation/cross-linking, as shown with the GII.4-specific antibodies such as human-derived NORO-320 and A1431 (Fig. 5a)[14,34]. Our studies show that the neutralization mechanism of M4 is distinct from either of these cases. As M4 binds away from the HBGA-binding site, it cannot directly block HBGA binding, and as the binding is monovalent, it cannot mediate cross-linking-specific aggregation blocking cell attachment (Fig. 5d). Surprisingly, we observed that M4 shares a considerable overlap with the non-neutralizing mAb A1227 which suggests that the larger size of A1227 results in increased steric clash and reduced binding to intact virions compared to M4 (Fig. 5b)[34].

When we model the M4 binding onto the recently published crystal structure of the GII.4 VLP[28], it is apparent that there is a significant steric clash between M4 and the neighboring VP1 dimers. It has recently been shown that GII.4 VP1 can be conformationally dynamic by adopting "resting" and "raised" conformations in the $T = 3$ capsid[28]. When the capsid adopts the stable "resting" conformation, M4 binding is sterically prohibitive (Fig. 4a–d). However, these steric clashes near

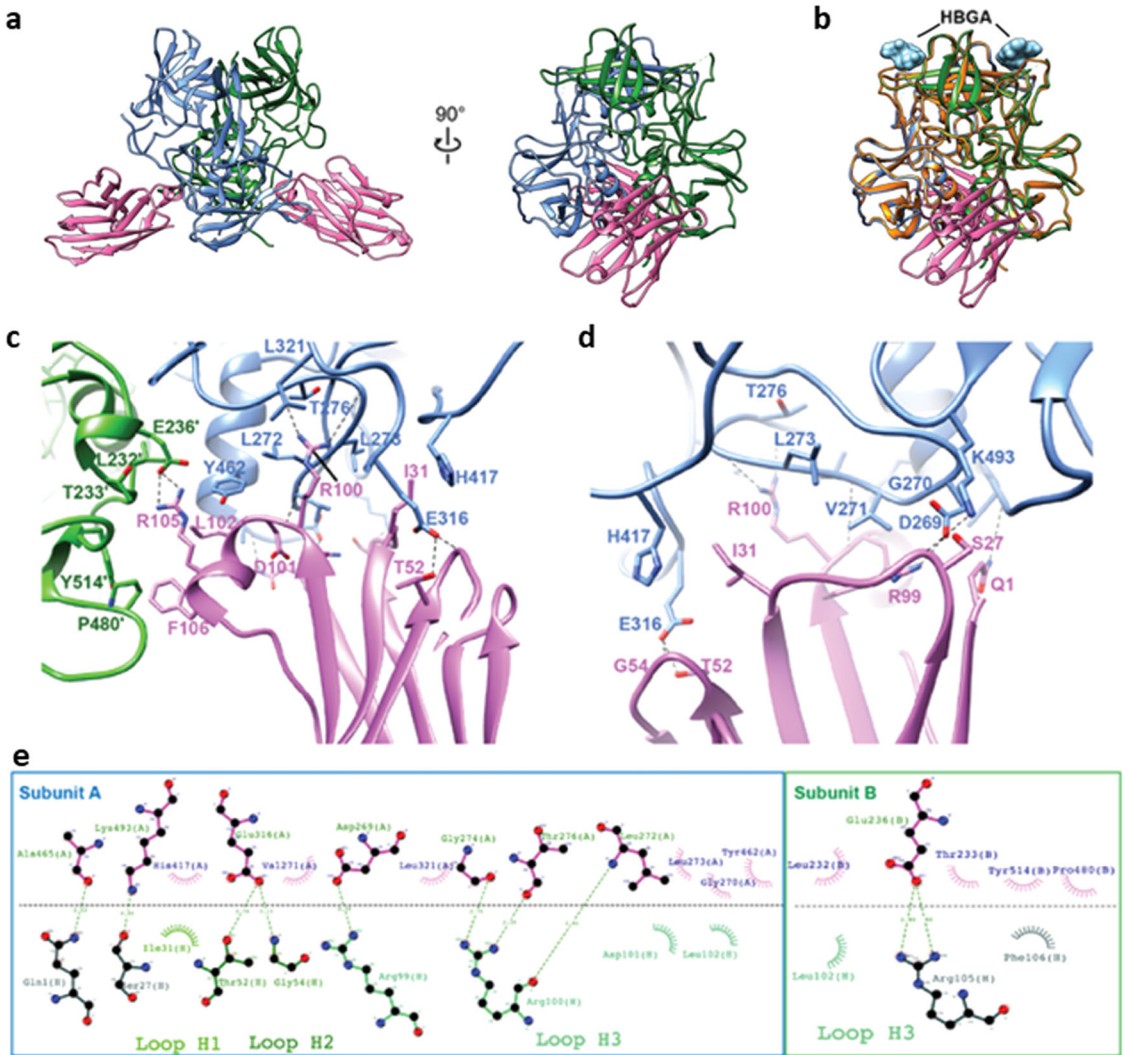

**Fig. 2 | Nanobody M4 in complex with GII.4 Sydney/2012 P-domain. a** X-ray crystal structure of the M4-GII.4 P-domain complex. The two subunits of the P-domain dimer are colored blue and green. Nanobody M4 is shown in pink. **b** Superposition of M4-GII.4 P-domain complex with HBGA-GII.4 P-domain structure (orange, PDB ID: 5J35). The glycan fucosyllactose (2′-FL) is shown as cyan spheres to indicate the HBGA-binding sites for reference. **c, d** Close-up views of the M4-binding site are lost when the P-domain breaks the stabilizing interactions of M4 with two subunits of P-domain dimer. The subunits A and B of the P-domain dimer are colored in blue and green, respectively. The side chains of M4 and P-domain are shown as stick representations. The hydrogen bonds are shown as black dashed lines. **e** Antibody plot analysis using the program LigPlot+ v.2.2.5. Green dashed lines indicate the hydrogen bonds, and short spokes radiating from each atom or residue represent the hydrophobic contacts.

the M4-binding site are lost when the P-domain breaks the stabilizing hydrogen bonds at the S-P-domain interface, rotates, and elevates above the shell domain transiting to the "raised" conformation (Fig. 4e–h). The kinetic equilibrium between the two states can be altered by variations in ionic strength or pH variations[21,28], allowing some VP1 subunits in the capsid to adopt "raised" conformation, thereby allowing access for M4 binding. M4 bound to the "raised" P-domain can act like a "lock" to prevent the P-domain from rotating back to rest on S-domain thereby restricting plasticity of the capsid and the particle becomes less stable as it loses several hydrogen-bonding contacts between the P and S domains (Fig. 4f). Considering that the binding affinity for M4 is high, M4 binding to several VP1 subunits can progressively destabilize the entire capsid. Indeed, when VLPs are treated with M4 and imaged using negative-stain EM, we observed particle disassembly into small debris (Fig. 4j). It is to be noted that M4-induced disassembly occurs even without the removal of bound cation, which has been shown necessary to trigger the transition from the resting to the raised state in the case of GII.4 VLP[28] and without changes in pH, which also may be a factor in triggering the conformational transition[21]. These results indicate M4 can readily alter the equilibrium between the resting and rising conformations and bind to its epitope regardless of external factors.

Further, recent work has shown that GII.4 VLPs and virions, but not P-domain alone, induce endosomal acidification to initiate endocytosis and uptake of the virus[17]. We performed similar endocytosis experiments with GII.4 VLPs in the presence of M4 and observed the endocytic uptake is significantly diminished, consistent with capsid disassembly by M4 as a possible mechanism of neutralization (Supplementary Fig. 5). Like M4, nanobody Nano-85 and monoclonal antibody 5B18, recognize an epitope near the S-P-domain occluded in the intact virion with VP1 adopting a resting conformation and disassemble the GII.4 VLPs (Fig. 5c)[35,38]. Although there have been no neutralization assays performed, Nano-85 and monoclonal antibody 5B18 are speculated to facilitate neutralization by disassembling the particles.

A remarkable observation from our studies is that M4 can neutralize multiple variants of GII.4 HuNoVs with potency significantly higher than other neutralizing antibodies reported to date. When compared to neutralization by NORO-320 mAb and Fab with $IC_{50}$ values of 11,690 ng/ml (33.4 nM) and 2950 ng/ml (59 nM)

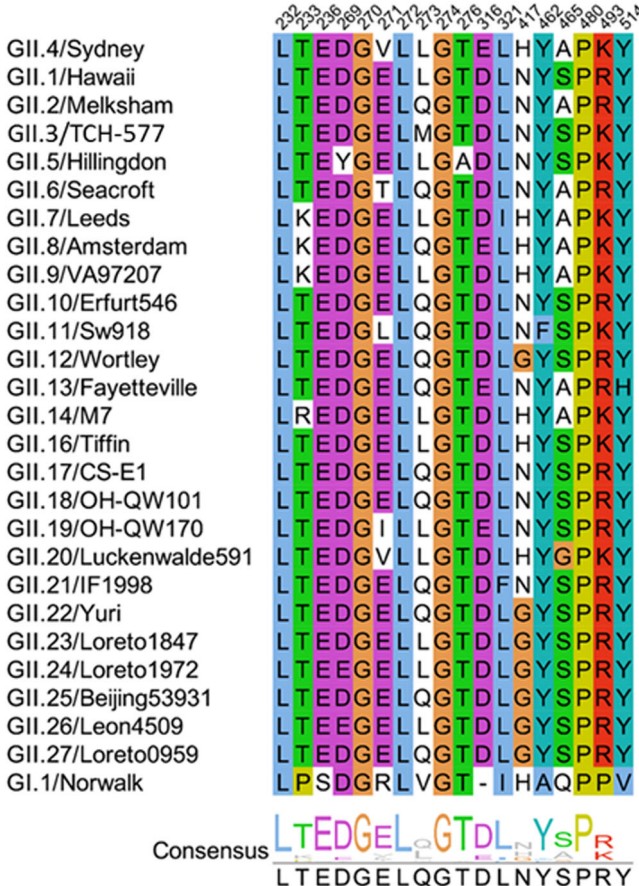

**Fig. 3 | Amino acid sequence alignment of the protruding domain.**
VP1 sequences of 26 GII genotypes and one GI.1 genotype are aligned using Clus-
talW and visualized by Jalview. The M4-binding residues are colored using ClustalX
shading scheme to highlight amino acid conservation.

respectively[14], M4 neutralizes GII.4 Sydney HuNoV with an IC$_{50}$ of
53 ng/ml (4 nM). The neutralization efficiency of M4 is also comparable
to 10E9 mAb (Fab fraction), which showed a GII.4 neutralization by
blocking HBGA binding with an IC$_{50}$ of 97 ng/ml (2 nM) in HIE[43]. The
better efficiency of M4 compared to NORO-320 may be due to its
smaller size to minimize steric clash and higher affinity of binding to
more readily engage its epitope to trigger disassembly.

An intriguing observation from our studies is that although M4
binds and disassembles GII.3 VLPs, similar to GII.4 VLPs (Fig. 4j), M4
only neutralizes GII.4 HuNoV and not GII.3 HuNoV (Fig. 1). This
observation suggests that M4 cannot readily access the epitope in the
GII.3 virions, perhaps because of significantly slower transition kinetics
between the "resting" and 'raising' states in the GII.3 virion compared
to that in the GII.3 or GII.4 VLPs, and also to the GII.4 virion. Although
the structures of $T=3$ GII.3[21] and GII.4 VLPs[28] are highly similar in their
resting conformations, GII.3 VLPs appear to exhibit slower kinetics of
transition between the resting and raised conformations. With changes
in pH or ion chelation, only 16% of the total GII.3 VLPs transit to the
raised P-domain conformation and the elevation of the P-domain in the
raised state is ~4 Å lower than the observed height for murine
norovirus[21]. In our infectivity assays, M4 fails to neutralize GII.3 even
after incubating the virus with M4 for an hour. In contrast, M4 dis-
assembles both GII.4 and GII.3 VLPs within 30 min of adding M4,
suggesting that the capsid structure in the GII.3 virion is intrinsically
more stable than that in the VLPs or in the GII.4 virion. It is plausible
that the genome along with VP2 in the GII.3 virion, contribute differ-
entially to the increased stability of the capsid. Our studies thus
underscore, despite a conserved epitope, how the capsid stability and

plasticity variations between genotypes may influence the mechanism
of neutralization.

In summary, our studies presented here provide insight into a
plausible novel mechanism of a nanobody that neutralizes multiple
GII.4 NoV variants with high potency by targeting a highly conserved
epitope that is vulnerable to inherent conformational dynamics of the
viral capsid. Given the advantages associated with nanobodies, these
studies could be helpful in further optimization of nanobody scaffolds
as efficient immunotherapeutic agents for periodically evolving GII.4,
and possibly other GII HuNoVs.

## Methods

### Virus neutralization assay
Human jejunal intestinal enteroids (J4$^{FUT2}$ HIEs[58]) were plated and
differentiated as HIE monolayers in collagen IV-coated 96-well plates
in commercial Intesticult human organoid growth medium (StemCell
Technologies, Cat#06010) as previously described[59]. Separately, 100
TCID$_{50}$ of GII.4 (Sydney 2012), GII.4 (Den Haag), GII.4 (New Orleans),
and GII.3 (TCH04-577) stool filtrates were preincubated, in Intesticult
differentiation medium supplemented with 500 μM of bile acid gly-
cochenodeoxycholic acid (GCDCA), with different concentrations of
M4 nanobody for 1 hr at 37 °C. Virus-nanobody mixtures were then
added to HIE monolayers and incubated at 37 °C. After 1 hr incuba-
tion, the inoculum was removed, and monolayers were washed twice
with CMGF [–] medium to remove unbound virus. Intesticult differ-
entiation medium (100 μl containing 500 μM GCDCA) was then
added to each well, and the cultures were incubated at 37 °C for 24 h.
Neutralization in each case was evaluated in two independent assays
for each genogroup/variant and samples were run in triplicate in
each assay.

Total RNA was isolated from each infected well using the King-
Fisher Flex purification system in conjunction with the MagMAX-96
viral RNA isolation kit. RNA extracted at 1 hpi served as the reference
point to assess the remaining quantity of viral material associated
with cells after washing the infected cultures. For viral quantification,
reverse transcriptase quantitative PCR (RT-qPCR) was performed
with the QNIF2d/COG2R/QNIFS primer pair and probe. The reaction
was conducted using the qScript XLT One-Step RT-qPCR ToughMix
reagent with ROX reference dye (Quanta Biosciences). Amplification
was performed on an Applied Biosystems StepOnePlus thermocycler
using the following cycling conditions: 50 °C (15 min) and 95 °C
(5 min), followed by 40 cycles of 95 °C (15 sec) and 60 °C (35 sec). A
standard curve based on a recombinant HuNoV GII.4 RNA transcript
was established to quantify viral genome equivalents (GEs) in RNA
samples. Viral replication was determined by RNA levels quantified
from samples extracted at 24 hpi. Percent reduction in GEs relative to
medium (100%) was determined from a total of 6 or 12 readings,
comprising biological replicates and technical duplicates to ensure
reliability, rigor, and robustness in our data analysis.

### Expression and purification of HuNoV VLPs
The VP1 and VP2 of GII.3 (Houston/TCH04-577/USA, AB365435) and
GII.4 (Sydney/2012/AUS, JX459908) noroviruses were expressed in the
baculovirus system as described previously[16,60]. Briefly, infected Sf9
cells were grown for 10 days, and the cell suspension was pelleted by
centrifugation at 22,100 × $g$ for 30 min. The resulting supernatant was
overlaid on a 30% sucrose cushion and VLPs were pelleted by cen-
trifuging at 120,000 × $g$ for 3 h at 4 °C. The pellet was suspended by
adding 200 μL of PBS per tube and incubating at 4 °C overnight. Sus-
pended VLPs were then pooled and diluted with PBS containing
cesium chloride (1.14 mg/ml) to a final concentration of 0.38 mg/ml
cesium chloride. The sample was then centrifuged at 150,000 × $g$ for
18 h at 4 °C. The VLP-containing white band was collected by micro-
pipette and then was dialyzed overnight at 4 °C in PBS pH 6.0. The
dialyzed VLPs were then further purified using a Sephacryl S500 size

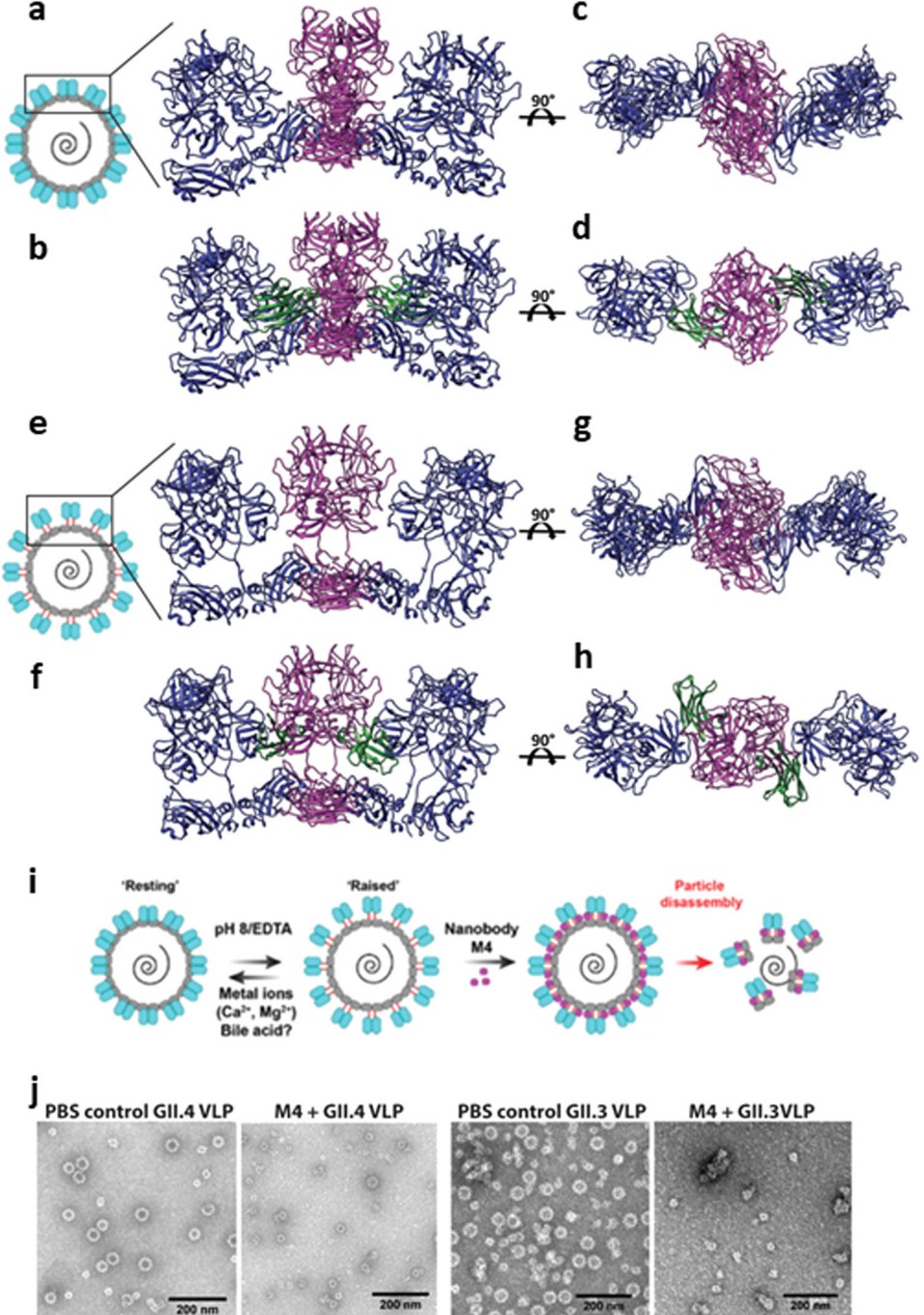

**Fig. 4 | Modeling of M4 bound on GII.4 capsid in the "resting" and "raised" states. a–d** Top view and side view of superimposition of M4-GII.4 P-domain complex onto the AB dimer of HOV VLP cryo-EM structure (PDB ID:7MRY) in "resting" state. **e–h** Top and side view of M4-GII.4 P-domain complex superimposed on the AB dimer of the VLP in the "raised" state, which was modeled in 8.0 Å GII.4c cryo-EM map in $T$ = 3 symmetry (EMD-10756). The VP1 subunits are colored (M4 bound dimer, magenta; neighboring dimer, blue). Nanobody M4 is shown in green. **i** Schematic of M4 neutralization mechanism. Cartoon representation of the capsid (designed using Adobe Illustrator) in the resting (as in **a**) and the raised state (as in **e**) shown with P-domain dimers in blue, S domain in green, and linker region in red. **j** Negative-stain EM analysis of GII.4 and GII.3 VLPs in the presence or absence of M4.

exclusion chromatography column. Purified fractions were pooled and stored at 4 °C.

### Expression and purification of P-domain and M4

Each of GII.3 (TCH04-577) and GII.4 Sydney 2012 P-domain sequence was cloned into the expression vector pMal-C2E (New England Bio-Labs). The recombinant P-domain was expressed with an N-terminal His6-maltose-binding protein (MBP) tag, and a tobacco etch virus (TEV) cleavage site between the MBP and P-domain in E. coli BL21(DE3)

and purified using His-Trap (GE Healthcare). The His-MBP tag was then removed using TEV protease and separated from the P-domain by purifying it once again using His-Trap (GE Healthcare), MBPTrap (GE Healthcare) affinity columns, and size exclusion chromatography as previously described[61]. The purified P-domain was concentrated and stored in a buffer containing 20 mM Tris-HCl (pH 7.2), 150 mM NaCl, and 2.5 mM MgCl$_2$. The recombinant M4 was expressed in *E. coli* WK6 strain. The periplasmic proteins were extracted by osmotic shock using Tris/EDTA/Sucrose (TES) buffer, and His-tagged M4 was purified

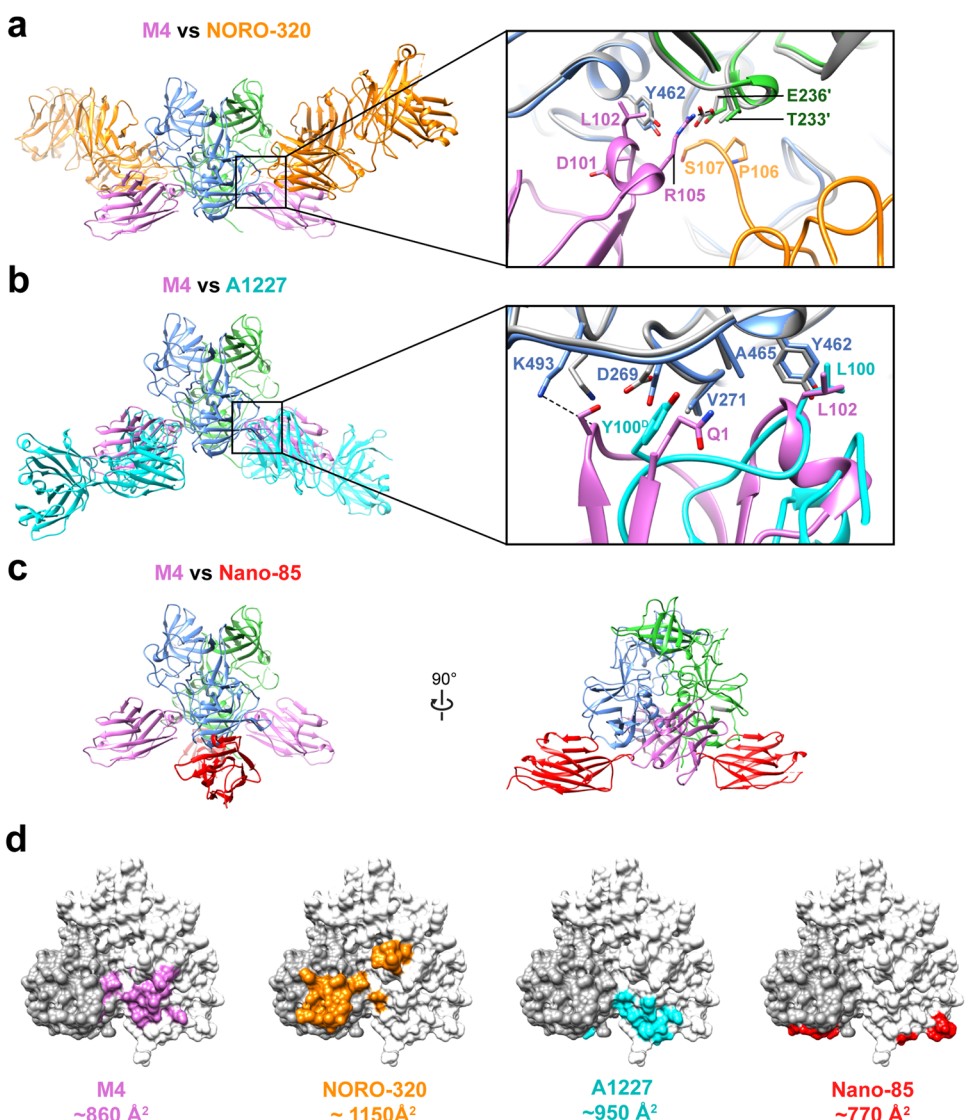

**Fig. 5 | Comparison of M4-GII.4 P-domain complex with representative GII.4-mAb or GII.4-nanobody complexes. a−c** Superimposition of the structure GII.4 P-domain in complex with M4 (pink, PDB ID: 8G0W), NORO-320 Fab (orange, PDB ID: 7JIE), A1227 Fab (cyan, PDB ID: 6N81), or nanobody Nano-85 (red, PDB ID: 4X7D). The subunits A or B of the P-domain dimer are colored in blue and green, respectively. The insets in (**a**) and (**b**) show the close-up view of the epitopes on the P-domain, with key side chains shown in the stick model and labeled. **d** The P-domain dimers are shown with white and dark gray surface representation to define each subunit, with the buried surface on P-domain colored as the corresponding nanobody or Fab.

from the periplasmic extract using a High-Trap HP Ni-chelating column (GE Healthcare, US).

### Crystallization of GII.4 P-domain and M4

Purified GII.4 P-domain and M4 were mixed with a 1:1.5 molar ratio and incubated for 1 h at 4 °C. The mixture was passed through an S200pg 16/60 gel filtration column, and the peak corresponding to the complex was collected. The size of the complex and the presence of both proteins were validated on an SDS-PAGE gel. The peak fractions were then pooled and concentrated to 10 mg/ml for crystallization trials. Crystallization screening using a hanging-drop vapor diffusion method at 20 °C was set up using a Mosquito nanoliter handling system (TTP LabTech) with commercially available crystal screens, and crystals were visualized using a Rock Imager (Formulatrix). The M4-GII.4 P-domain complex was crystallized in a buffer containing 2% v/v Tacsimate pH 4.0, 0.1 M Sodium acetate trihydrate pH 4.6, 16% w/v Polyethylene glycol 3350 (Hampton Research).

### Data collection and structure determination

Diffraction data were collected on beamline 8.2.2 at Advanced Light Source (Berkeley, CA) and processed using HKL2000[62]. The previously published GII.4 Sydney P-domain structure (PDB ID 7JIE) and the nanobody (PDB ID 5KW9) coordinates were used as the search models by molecular replacement using program PHASER[63]. Iterative refinement cycles and further model building were carried out using PHENIX[64] and COOT[65]. Data refinement and statistics are given in Table 1. The interactions between P-domain and the M4 were analyzed using LigPlot+ v.2.2.5[66]. Figures were prepared using Chimera[67].

### Negative-stain EM

GII.4 Sydney VLPs were diluted to a working concentration of 0.50 mg/ml in PBS pH 6.0. VLP was then mixed with PBS or M4 nanobody for a final concentration of 0.25 mg/ml (19.5 μM) of M4 and 0.25 mg/ml (4.23 μM) of VLP. Following the preparation of the mix, each condition was incubated for 30 min at room temperature. A 3 μL aliquot of each sample mixture was applied onto a glow-

discharged 200-mesh 2/2 Quantifoil holey carbon grid containing an 8 nm layer of carbon and incubated for 3 min. Grids were then blotted, washed with Milli-Q $H_2O$, and then 2% uranyl acetate was applied to the grids for 1 min. Finished grids were stored in a dehumidifier at room temperature. Images were collected at 120 kV on a JEM-1230 EM at a ×25,000 magnification.

## Biolayer Interferometry

Biotinylation of the GII.3 and GII.4 P-domains was carried out using EZ-Link NHC-LC-LC-biotin (catalog no. 21343; Thermo Scientific) following the manufacturer's instructions. The P-domain was loaded onto streptavidin biosensors at a concentration of 0.625 µg/ml in the BLI running buffer (20 mM HEPES buffer, 150 mM NaCl, 0.005% surfactant P20, 2 mg/ml bovine serum albumin, pH 7.8) for 300 s, resulting in capture levels of 0.8–1.5 nm within a row of eight tips. M4 was diluted in BLI running buffer to a final concentration of 20 nM and incubated on ice overnight. M4:P-domain association and dissociation curves were obtained through twofold serial dilutions of M4 (20, 10, 5, 2.5, 1.25, 0.625, 0.3125 nM) plus buffer blanks at 30 °C using the Octet acquisition software. BLI studies were carried out using an Octet RED96 instrument (FortéBio).

## Dynamic light scattering

The hydrodynamic diameters of GII.4 Sydney and GII.3 TCH-577 VLPs in the absence or presence of M4 at pH 6.0 were measured using DLS on a ZetaSizer Nano instrument (Malvern Instruments, U.K.). VLPs were diluted to a final concentration of 200 nM and M4 was diluted to 800 nM in phosphate-buffered saline (molar ratio 1:4 VP1:M4). Upon addition of M4, samples were incubated on ice for 30 min before measurements were taken. Three × 12 measurement runs were performed with standard settings (Refractive Index 1.335, viscosity 0.9, temperature 25 °C).

## Sedimentation velocity-analytical ultracentrifugation

The SV-AUC experiments with VLPs of GI.1 Norwalk, GII.4 Sydney, and GII.3 TCH-577, both alone and in complex with M4, were performed using Beckman-Coulter XL-A analytical ultracentrifuge with a TiAn60 four-hole rotor and two-channel Epon centerpieces (12 mm). VLPs in complex with M4 were prepared at a molar ratio of 1:4 with working concentrations of 1 µM VP1 and 4 µM M4, in PBS at pH 6.0, and incubated at 37 °C for 1 h. GI.1 VLPs treated with M4 nanobody were used as negative control. Absorbance scans were recorded at 280 nm at every 1 min interval at 1500 rpm at 4 °C. Continuous distribution c(s) model was used to fit multiple scans at regular intervals with SEDFIT[68,69]. The solvent density ($\rho$) and viscosity ($\eta$) were calculated from the chemical composition of different proteins by SEDNTERP[69].

## Endocytosis assay

Endocytosis measurements were carried out using FM1-43FX (ThermoFisher Scientific) as described previously[17]. Briefly, HIE monolayers were treated with 10 µg/ml of FM1-43FX for 10 min at 37 °C with either VLP alone or VLP preincubated with M4 for 1 h at 37 °C. Monolayers were washed with prechilled PBS and fixed in 4% PFA for 20 min. followed by nuclei staining with 300 nM DAPI for 5 min at room temperature. Endocytic trafficking was measured by the presence of fluorescent puncta which were observed by epifluorescence microscopy using Olympus cellSens Standard Version 2.3 software. Quantitation of the number of fluorescent puncta was done using J-image. Every experiment was repeated at least three times with 4 images analyzed per condition in each experiment.

## Statistics and reproducibility

Each experiment was repeated independently at least twice with similar results and representative data were shown.

## Reporting summary

Further information on research design is available in the Nature Portfolio Reporting Summary linked to this article.

## Data availability

Atomic coordinates and structure factors for the crystal structure of M4 in complex with GII.4 P-domain have been deposited in the Protein Data Bank under the accession code 8G0W. The authors declare that all other data supporting the findings of this study are available within the paper and its supplementary information files. Source data are provided with this paper.

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

## Acknowledgements

We acknowledge the Advanced Light Source (8.2.2) (Berkeley, CA) for X-ray data collection. The ALS-ENABLE beamlines are supported in part by the National Institutes of Health, National Institute of General Medical Sciences, grant P30 GM124169-01. The Advanced Light Source is a Department of Energy Office of Science User Facility under Contract No. DE-AC02-05CH11231.This work was supported by N.I.H. grant P01 AI057788 (to M.K.E, R.L.A, and B.V.V.P.) and a grant from the Robert Welch Foundation (to B.V.V.P.). P30 CA125123 supported the Protein and Monoclonal Antibody Production Shared Resource at Baylor College of Medicine for VLP expression and SV-AUC. We also thank Fulbright Program for a grant for Dr. Bok to travel to NIH for HIE experiments. This work was also supported, in part, by the Division of Intramural Research, NIAID, NIH.

## Author contributions

W.S., L.H., B.V.V.P., and M.K.E. conceived and designed the research in collaboration with V.P., M.B., E.K., S.V.S., and K.G. who provided the M4 nanobody. E.K. performed neutralization assays, W.S. expressed and purified the P-domain and crystallized it in complex with M4. W.S. L.H. and B.S. performed data collection and structure determination and analyzed the structure with B.V.V.P. W.S. and S.K. performed DLS and SV-AUC experiments, and A.V. performed endocytosis assay. F.H.N, R.L.A., and S.S. provided reagents. All authors reviewed, edited, and approved the final manuscript.

## Competing interests

M.K.E. is named as an inventor on patents related to cloning of the Norwalk virus genome, is a consultant to Takeda Vaccines, Inc., and has received support from Takeda Vaccines, Inc. R.L.A. has received support from Takeda Vaccines, Inc. The remaining authors declare no competing interests.
