## [Peer Review File · Nature Communications]

nature portfolio

Peer Review FileReviewer comments, first round:

Reviewer #1 (Remarks to the Author):

This study examined the neutralisation potential of a nanobody (M4) using a norovirus cell culture system (with human intestinal enteroids). The M4 nanobody was able to neutralise GII.4 norovirus with a high potency in cell culture but was unable to inhibit GII.3 noroviruses in the same system. The M4 nanobody binding site was determined using X-ray crystallography and the binding site was located on the bottom of the P1 subdomain. Dynamic light scattering (DLS) suggested that M4 nanobody engagement with norovirus VLPs resulted in complexed particles with a decrease in particle stability and was suggested to trigger disassembly of the VLPs. Structural modelling on intact VLP structures (with two forms of the P domains: raised or resting on the shell domain) suggested that the M4 nanobody bound to P domains when they were raised off the shell domain. The conclusions are not completely novel and have been shown before for norovirus capsid-specific monoclonal antibodies and nanobodies.

1. Shanker S, et al. Structural basis for norovirus neutralization by an HBGA blocking human IgA antibody. *Proc Natl Acad Sci U S A*. 2016;113(40):E5830-E7.
2. Costantini V, et al. Human Norovirus Replication in Human Intestinal Enteroids as Model to Evaluate Virus Inactivation. *Emerg Infect Dis*. 2018;24(8):1453-64.
3. Alvarado G, et al. Broadly cross-reactive human antibodies that inhibit genogroup I and II noroviruses. *Nat Commun*. 2021;12(1):4320.
4. Koromyslova AD, et al. Nanobodies targeting norovirus capsid reveal functional epitopes and potential mechanisms of neutralization. *PLoS Pathog*. 2017;13(11):e1006636.
5. Koromyslova AD, et al. Nanobody-Mediated Neutralization Reveals an Achilles Heel for Norovirus. *J Virol*. 2020;94(13).
6. Ruoff K, et al. Structural Basis of Nanobodies Targeting the Prototype Norovirus. *J Virol*. 2019;93(6):e02005-18.
7. Koromyslova AD, et al. Nanobody binding to a conserved epitope promotes norovirus particle disassembly. *J Virol*. 2015;89(5):2718-30.
8. Kher G, et al. Direct Blockade of the Norovirus Histo-Blood Group Antigen Binding Pocket by Nanobodies. *J Virol*. 2023;10.1128/jvi.01833-22:e0183322.

Major concerns:

The work will be important for norovirus therapeutic development but does not answer the question as to how the M4 nanobody causes particle disassembly.

The speculations of the disassembly are lacking convincing evidence in several experiments. For example, the EM images (Fig. 4d) appear to show different sized GII.4 particles (T=1, T=3, T=4) in the untreated samples, and in the M4 nanobody treated samples there are still GII.4 particles, which appear to be T=1, T=3. Moreover, the GII.3 VLPs are not homogenous in size (Fig. 4d). This would make analysis of treatment difficult. The author mentioned that M4 binds GII.1,2,3,4,6,7 genotypes. Where these tested in the cell culture neutralization assays? It would be great to see how M4 nanobody binds to these genotypes (with modelling or X-ray structure), but doesn't block in cell culture (i.e., GII.3). Similarly, this could answer what needs to be done to improve the neutralisation.

The modelling of M4 nanobody binding to the resting or raised VLP is pure speculation (Fig. 4c). There is no proof that M4 binds to either of these T=3 states, as the EM images are mixed with different sized particles. Can M4 nanobodies bind to T=1 particles? Are virions raised or lowered on the shell domain? If M4 binds raised VLPs, then virions are also raised? Could you discuss or test this somehow?

Can the authors explain the massive standard deviations with the controls, and several treatments in Figure 1?

Can you explain why the EM images of untreated VLPs (Fig. 4d) are different sizes including T=1

(~25 nm), but in the DLS (Supp Fig 1) all appear 45 nm? The untreated VLPs would not appear to be comparable to native virions.

Why did the authors measure the VLP diameters using % volume?

Was DLS performed at 37C to see if there was any difference and different treatment times?

Did the authors consider any alternative binding orientations in the crystallography structures and mutagenesis to confirm the binding at the predicted site?

Where HBGA blocking experiments done with these VLPs and is it possible to block virions using the HBGA blocking assay?

Reviewer #2 (Remarks to the Author):

In the present manuscript Salmen and colleagues use X-ray crystallography augmented by a suite of other experiments to elucidate the structural basis by which nanobody M4 neutralizes several norovirus variants. Interestingly, the authors show that M4 neutralizes GII.4 but not GII.3 strains. The structural work is of very high quality and I consider the study to be important for virologists. However, I believe that comparing M4 with another nanobody from the ones characterized in their previous study, would have made a much stronger article.

Overall, the results are very interesting and presented with solid evidence. However, the authors should try to improve some aspects of the manuscript and put more effort in the discussion. Here are few points that I suggest to be considered

Introduction

- The authors mention a series of studies on nanobodies against other viruses (lines 82-84). However, they insist on enveloped viruses, while many articles on non-enveloped viruses are also abundant (i.e. poliovirus, by Hogle group) and their mechanism of action is more relevant to the current study.
- The author should mention the cause of the rising and resting position of the P domain, for instance influenced by pH (Lines 68-69) as shown by Smith group.
- Consider mentioning lines 250-253 in the introduction after lines 68-69
- The authors could explain why they chose only M4 from all the nanobodies discussed in their 2015 article.

Methods, Results

- The results section should mention the pH at which the data was collected
- Figure 4a,b could be improved to show how M4 sits between the dimers. (as for instance in a Murat group article from 2020)

Discussion

- Several places in the discussions could be better supported by references. For instance, lines 224-226. The authors should show the structure of GII.4 Sydney at pH 6 without M4 to clearly see if it is initially in resting or rising state. Also, please explain if GII.4 Sydney transitions from resting to rising state in response to pH; or does it naturally exist in rising state?
- How is the capsid structure (P domain orientation) of GII.3 different from GII.4 Sydney? Past study showed that only 16% exist in rising conformations. Could the orientation of the P domain of GII.3 in resting state allow M4 to bind?
- Check again carefully all the references! For instance, in reference 6, Mortality is certainly not the first author.

Reviewer #3 (Remarks to the Author):

The manuscript by Salmen et al entitled 'A single nanobody neutralizes multiple epochally evolving

human noroviruses by modulating capsid plasticity' presents X-ray structure of human norovirus GII.4 Sydney P-domain (part of the viral protein 1, VP1) in complex with a neutralizing llama-derived nanobody M4. The structure reveals a conserved epitope among GII.4 HuNoVs targeted by M4. Comparison of these results with the structure of the capsid (of viral like particles, VLPs) indicates that M4 binds to the raised VP1 conformation (GII.4 VP1 is dynamic and can adopt 'resting' and 'raised' conformations in the capsid). This is an important information as up to now most of the GII HuNoV neutralizing antibodies have been shown to block receptor binding. By additional assays authors have shown that M4 binding to the capsids leads to capsid dissociation. Nanobodies like M4 potentially could be further developed as antivirals against GII.4 noroviruses.

Major points

1. Authors should check they X-ray derived atomic model. Zero Ramachandran outliers do not fit well with the presence of other outliers (like RSRZ outliers) and the reported resolution 2.87 Å; in addition, reported Rfree is relatively high meaning that there is some discrepancy between the model and the experimental data. Authors should try to reduce Rfree even at the expense of few Ramachandran outliers. Zero Ramachandran outliers can be misleading (<https://doi.org/10.1016/j.str.2020.08.005>).
2. Description of endocytosis assay is insufficient, authors should revise.

Specific comments

The title

Authors use a term 'multiple epochally evolving human noroviruses' in the title. It would be useful to explain for the reader what does in mean, for example in the introduction.

Lines 117-120, 'A superposition of the structures of GII.4 Sydney P-domain alone and in complex with M4 and in complex with H-type HBGA showed that M4 binding does not result in any significant conformational changes in the P-domain, with a Ca RMSD of 0.68 Å, and HBGA binding site is not affected (Fig. 2b).' Not clear for which structure comparison RMSD is shown, P-domain alone vs P-domain in complex with M4 or P-domain alone vs P-domain in complex with HBGA. Fig 2 Grey is hard to distinguish from blue.

Line 73: It would be good to know more on histo-blood group antigens (HBGAs), at least from the structural point of view.

Line 125: It is not clear what are P1 and P2 subdomains. Some schematics could help.

Line 139: 'shows that the identified M4 binding site in our crystallographic structure is highly conserved (72% to 94% similarity)' How the amino acid similarity is defined? If it is identity, then word 'identity' should be used.

Figure 4 a and b, the M4 binding site is not seen in these figures. In addition to showing atomic structures inside the cryoEM density, authors should include visual for atomic models of P with/without M4 in context of the virion (based on the VLP structures) with highlighted (clearly visible) binding site for the nanobody M4. Perhaps different angle/view could help here.

Lines 229-231: 'We performed similar endocytosis experiments with GII.4 VLPs in the presence of M4 and observed the endocytic uptake is significantly diminished, consistent with capsid disassembly by M4 as a mechanism of neutralization (Sup. Fig. S3).' This experiment is not described in the results.

Discussion section looks like a mixture of results and discussion. Having results as a separate section, all results should be described in results section.

Line 284: 'reverse transcriptase aunitative PCR (RTqPCR)' should be 'reverse transcriptase quantitative PCR (RTqPCR)'.

Line 337: please add final molar concentrations for both VLPs and M4 to indicate molar ratio of nanobody vs its target (VLPs).

Lines 362-369: Description of endocytosis assay is insufficient. Please clarify what is FM1-43FX (ThermoFisher Scientific). Line 365: 'Briefly, HIE monolayers were 10 µg/mL of FM1-43FX for 10 min at 37 °C with either VLP alone or VLP pre-incubated with M4 for 1 h at 37 °C.' From this sentence it is not clear what has been done with HIE monolayers. It is not clear how authors measured fluorescence before quantitation.

The quality of the structure (PDB report): the atomic model shows zero (0) Ramachandran outliers

at the resolution of 2.87 Å (somewhat low resolution for X-ray structures). The reported Rfree is somewhat high. Authors should try to improve their model by reducing Rfree even if it will introduce few Ramachandran outliers. Zero Ramachandran outliers can be misleading; authors should consult for example this paper <https://doi.org/10.1016/j.str.2020.08.005>.

Nature Communications manuscript NCOMMS-23-11638

We thank all the reviewers for their positive comments and suggestions. We have addressed each of the reviewer's comments in detail and revised the text accordingly. Our responses are in blue.

Reviewer #1 (Remarks to the Author):

This study examined the neutralization potential of a nanobody (M4) using a norovirus cell culture system (with human intestinal enteroids). The M4 nanobody was able to neutralize GII.4 norovirus with a high potency in cell culture but was unable to inhibit GII.3 noroviruses in the same system. The M4 nanobody binding site was determined using X-ray crystallography and the binding site was located on the bottom of the P1 subdomain. Dynamic light scattering (DLS) suggested that M4 nanobody engagement with norovirus VLPs resulted in complexed particles with a decrease in particle stability and was suggested to trigger disassembly of the VLPs. Structural modelling on intact VLP structures (with two forms of the P domains: raised or resting on the shell domain) suggested that the M4 nanobody bound to P domains when they were raised off the shell domain.

The conclusions are not completely novel and have been shown before for norovirus capsid-specific monoclonal antibodies and nanobodies.

1. Shanker S, et al. Structural basis for norovirus neutralization by an HBGA blocking human IgA antibody. *Proc Natl Acad Sci U S A*. 2016;113(40):E5830-E7.
2. Costantini V, et al. Human Norovirus Replication in Human Intestinal Enteroids as Model to Evaluate Virus Inactivation. *Emerg Infect Dis*. 2018;24(8):1453-64. (cultivation)
3. Alvarado G, et al. Broadly cross-reactive human antibodies that inhibit genogroup I and II noroviruses. *Nat Commun*. 2021;12(1):4320.
4. Koromyslova AD, et al. Nanobodies targeting norovirus capsid reveal functional epitopes and potential mechanisms of neutralization. *PLoS Pathog*. 2017;13(11):e1006636.
5. Koromyslova AD, et al. Nanobody-Mediated Neutralization Reveals an Achilles Heel for Norovirus. *J Virol*. 2020;94(13).
6. Ruoff K, et al. Structural Basis of Nanobodies Targeting the Prototype Norovirus. *J Virol*. 2019;93(6):e02005-18.
7. Koromyslova AD, et al. Nanobody binding to a conserved epitope promotes norovirus particle disassembly. *J Virol*. 2015;89(5):2718-30.
8. Kher G, et al. Direct Blockade of the Norovirus Histo-Blood Group Antigen Binding Pocket by Nanobodies. *J Virol*. 2023;10.1128/jvi.01833-22:e0183322.

Thank you for the comments. Although structures of norovirus P domain in complex with various antibodies have been published, including those from our lab, what distinguishes our current study from these is that it is the **first to demonstrate** that a single nanobody neutralizes infection of multiple GII.4 epidemic variants using HIEs, accompanied by a structural description with mechanistic implications, and providing further validation for the neutralization mechanism using endocytic entry experiments in HIEs. We have made this important distinction abundantly clear in the title and the text.

Major concerns:

- 1) The work will be important for norovirus therapeutic development but does not answer the question as to how the M4 nanobody causes particle disassembly.

The speculations of the disassembly are lacking convincing evidence in several experiments. For example, the EM images (Fig. 4d) appear to show different sized GII.4 particles (T=1, T=3, T=4) in the untreated samples, and in the M4 nanobody treated samples there are still GII.4 particles, which appear to be T=1, T=3. Moreover, the GII.3 VLPs are not homogenous in size (Fig. 4d). This would make analysis of treatment difficult.

In addition to EM and DLS analyses, in the revised manuscript we have added new data from sedimentation-velocity analytical ultracentrifugation (SV-AUC), which provides further strong evidence for M4-triggered disassembly of GII.4 and GII.3 VLPs. While DLS indicates changes in particle dimensions, SV-AUC indicates changes in molecular weight. In addition, we have added GI.1. Norwalk virus VLPs as a negative control for both DLS (Supp Fig. 2) and SV-AUC (Supp Fig. 3) experiments. All these data **unequivocally demonstrate** that M4 disassembles GII.4 and GII.3 VLPs irrespective of the heterogeneity in their sizes or T numbers. As can be seen in the Supp. Fig. 2, and Supp. Fig. 3 that upon M4 treatment, there is a significant shift toward reduced sizes (as shown by DLS) and molecular weights (as shown by SV-AUC) of GII.4 and GII.3, whereas GI.1 VLPs, used as a negative control, as the epitope for M4 is not conserved, remained unchanged. See Supplemental Fig. 2 and Fig. 3.

2) The author mentioned that M4 binds GII.1,2,3,4,6,7 genotypes. Were these tested in the cell culture neutralization assays?

We have shown neutralization experiments for GII.3. Although we have cultivated GII.1, GII.2 or GII.7. HuNoVs, we were unable to perform neutralization assays with these viruses because we did not have sufficient stool of high enough titer to do these assays.

3) It would be great to see how M4 nanobody binds to these genotypes (with modelling or X-ray structure), but doesn't block in cell culture (i.e., GII.3). Similarly, this could answer what needs to be done to improve neutralization.

As mentioned in the paper, due to the high degree of sequence conservation of the epitope, we predicted GII.3 as well as these other GII strains will bind similarly. We have clearly shown using biolayer interferometry (see Supplemental Fig. 4) that M4 binds GII.3 P-domain with a similar high affinity as it does to the GII.4 P-domain.

GII.3 capsid in the resting stage is very similar to GII.4 structure, and modeling M4 into GII.3 capsid structure shows similar steric hindrance as in GII.4 capsid, which is relieved only upon transition to raised conformation. As we have noted in the discussion, the likely reason is the difference in transition kinetics between GII.3 and GII.4 VLPs, and plausibly the GII.3 virions are more stable than GII.4 or GII.3 VLPs.

4) The modeling of M4 nanobody binding to the resting or raised VLP is pure speculation (Fig. 4c). There is no proof that M4 binds to either of these T=3 states, as the EM images are mixed with different-sized particles. Can M4 nanobodies bind to T=1 particles? Are virions raised or lowered on the shell domain? If M4 binds raised VLPs, then virions are also raised. Could you discuss or test this somehow?

As noted above, irrespective of the T numbers that may be there in the sample, there is a clear decrease in the overall distribution of the particle sizes (by DLS) and molecular weights (by SV-AUC), which would

not have happened if M4 did not bind. For instance, in our negative control, the M4 treatment did not change anything as it does not bind GI.1. *Whether M4 binds to T=1 particles is not our focus, instead our focus is what happens when M4 binds to T=3 particles, as T=3 symmetry is what is expected in the virions.* With molecular modeling of the M4 binding in the context of T=3 capsid structure, not only can we understand why the disassembly occurs, but it clearly shows that M4 cannot bind if VP1 is in the resting conformation as it sterically clashes with the neighboring subunits in the capsid. In contrast, in the raised conformation, these contacts are relieved. But the raised conformation, with fewer hydrogen bonds and less interaction surface between P and the S domains, is less stable than in the closed state with more hydrogen bonds and more interacting surface; as a result, with M4 binding, the capsid becomes increasingly unstable leading to disassembly as we have explained in the manuscript.

Regarding the question “Are virions raised or lowered on the shell domain”, please note that the S domain is not altered, it is only the P domain that rotates and raises above the S domain, and the S domain stays at the same location.

5) Can the authors explain the massive standard deviations with the controls and several treatments in Figure 1?

We appreciate and thank the reviewer for this insightful comment. The variability observed (large standard deviations) for some of the data in Figure 1 is associated with many factors. They include the decreasing (or partial) inhibition observed as the nanobody concentration decreases, in some instances, the lack of inhibition at the concentration tested, differing replication characteristics of the infecting strain, and the variability present in the replication system in general. The variability does not affect the overall conclusions of the data shown, where clear gradients of inhibition by the nanobody are observed and an IC50 can be calculated.

6) Can you explain why the EM images of untreated VLPs (Fig. 4d) are different sizes including T=1 (~25 nm), but in the DLS (Supp Fig 1) all appear 45 nm? The untreated VLPs would not appear to be comparable to native virions.

DLS provides approximate distributions of particle diameters in solution represented by a bell curve accounting for variance in the diameter of particles. As can be seen, compared to the untreated VLPs, upon M4 treatment, the curve shifts toward a distribution that corresponds to particles of significantly smaller diameters. These could be T=1 particles or aggregates of disassembled products. Based on this reviewer’s comment, we performed SV-AUC to provide more quantifiable data (see the response to point 1 above) of VLPs before and after M4 treatment. This analysis is also consistent with EM and DLS showing that M4 binding to VLPs destabilizes the GII.4 and GII.3 VLPs. This is further unequivocally supported by our **negative control using GI.1 VLPs**, whose particle size distribution remains the same before and after M4 treatment in both DLS and AUC. This is because the epitope for M4 is not conserved in GI.1.

We have included additional DLS data and new AUC data on GII.4, GII.3 and GI.1 VLPs before and after M4 treatment in Supplementary Figs 2 and 3. Based on this new data, we have revised the section “M4 induces particle disassembly of GII.4 and GII.3 VLPs with different kinetics”. Lines 161-187,

Why did the authors measure the VLP diameters using % volume?

When analyzing DLS data, the Malvern zetasizer software calculates a correlation function based on Brownian motion of the particles in the solution. This correlation function can then be used to calculate the size distribution of particles. Three primary types of size distributions can be generated based on the correlation decay rate; intensity, volume, and number distributions. Using intensity distribution biases the distribution to higher-order components, such as aggregates, because large particles scatter much more light than smaller particles proportional to the sixth power of its diameter (Rayleigh's approximation). In the case of volume distribution, the calculation is based on the volume of a sphere ($\frac{4}{3}\pi(r)^3$) which more evenly measures a range of particle sizes. Therefore, in the DLS analysis, we used volume distribution to focus on smaller order populations, such as the 25-45 nm populations. We have clarified the analysis in the figure. **Irrespective of the choice of how the DLS data is presented, the conclusion remains unequivocally the same, M4 destabilizes the GII.4 and GII.3 VLPs but not GI.1 VLPs.**

Was DLS performed at 37C to see if there was any difference and different treatment times?

Yes, we performed DLS at 37C and did not see any difference. Based on neutralization studies, we observed neutralization of GII.4 at 1-hour post-treatment. We, therefore, decided to perform DLS of both VLPs following 1-hour incubation with M4 to check for changes in particle stability.

Did the authors consider any alternative binding orientations in the crystallography structures and mutagenesis to confirm the binding at the predicted site?

The binding is not predicted, it is determined using X-ray crystallography, and it is of **nM affinity** as shown using BLI. Our crystallographic structure of the P domain in complex with M4 is determined at a high enough resolution with excellent data quality and refinement statistics to unequivocally identify the epitope-paratope interactions. In our crystal structure, we did not see any alternate binding of M4. With such strong binding affinity and crystallographic data, we do not see any necessity for mutagenesis to confirm the epitope for M4.

Where HBGA blocking experiments done with these VLPs and is it possible to block virions using the HBGA blocking assay?

HBGA binding has been performed using M4 with GII.4 (MD145) and GI.1 (Norwalk) VLPs in our previous publication (Garaicoechea, L. et. al. 2015), as was noted in the original version of the manuscript. It is currently not feasible to perform a quantitative HBGA blocking assay with authentic virions due to the lack of robust cultivation systems. The only source for the virions is the stool specimen.

Reviewer #2 (Remarks to the Author):

In the present manuscript Salmen and colleagues use X-ray crystallography augmented by a suite of other experiments to elucidate the structural basis by which nanobody M4 neutralizes several norovirus variants. Interestingly, the authors show that M4 neutralizes GII.4 but not GII.3 strains.

The structural work is of very high quality and I consider the study to be important for virologists. However, I believe that comparing M4 with another nanobody from the ones characterized in their previous study, would have made a much stronger article.

Overall, the results are very interesting and presented with solid evidence. However, the authors should try to improve some aspects of the manuscript and put more effort in the discussion. Here are few points that I suggest to be considered

Introduction

- The authors mention a series of studies on nanobodies against other viruses (lines 82-84). However, they insist on enveloped viruses, while many articles on non-enveloped viruses are also abundant (i.e. poliovirus, by Hogle group) and their mechanism of action is more relevant to the current study.

Thank you for the suggestion, we have included articles referencing nanobody studies about non-enveloped viruses. Updated Lines 85-88.

- The author should mention the cause of the rising and resting position of the P domain, for instance influenced by pH (Lines 68-69) as shown by Smith group.

Thank you for the comment, we elaborated on the primary driver of the raised position in GII.4 VLPs is the removal of stabilizing ions, and pH in the case of GII.3 VLPs including referencing papers from the Smith group in Lines 70-75

- Consider mentioning lines 250-253 in the introduction after lines 68-69

Thank you for the suggestions. We have done this in the revised version. Lines 73-74

- The authors could explain why they chose only M4 from all the nanobodies discussed in their 2015 article.

Thank you for the comment. As mentioned in the introduction, we investigated M4 because it showed high cross-reactivity to several GII.4 variants and blocked HBGA binding in surrogate studies. Therefore, we decided it had the highest potential for therapeutic development and basic biology research. We have revised the text to clarify this point (lines 88-92).

Methods, Results

- The results section should mention the pH at which the data was collected
- Figure 4a,b could be improved to show how M4 sits between the dimers. (as for instance, in a Murat group article from 2020)

Thank you for the suggestion. As suggested, we have modified Figures 4a and 4b and reworded the legend accordingly. We have also included the pH at which the data were collected.

Discussion

- Several places in the discussions could be better supported by references. For instance, lines 224-226. The authors should show the structure of GII.4 Sydney at pH 6 without M4 to clearly see if it is initially in

a resting or rising state. Also, please explain if GII.4 Sydney transitions from resting to rising state in response to pH; or does it naturally exist in a rising state?

We have revised this portion of the text corresponding to lines 224-226 in the original version and provided the refs; lines 244-245 in the revised version.

In the revised Fig.4, we have clearly shown the GII.4 VLP structure with M4 in both resting and rising conformations. As we have demonstrated previously (Hu et al., 2022, ref 28), the transition from a stable resting to raised state occurs only upon chelating the bound ion by EDTA; variations in pH do not trigger this transition. In GII.3 VLP, however, a higher pH triggers the transition (Song et al. 2020, ref 21) We have clearly stated this in the revised manuscript with refs; see lines 73-75, and 252-255.

- How is the capsid structure (P domain orientation) of GII.3 different from GII.4 Sydney? A past study showed that only 16% exist in rising conformations. Could the orientation of the P domain of GII.3 in the resting state allow M4 to bind?

When we compare the 9.3 Å cryo-EM map for T=3 GII.3 in the resting state (Song et. al. 2020, ref 21) with our published GII.4 VLP structure (Hu et al., 2022, ref 28), also in the resting stage, the overall capsid structure and the P domain orientation are very much the same. This indicates that in both GII.4 VLP (see revised Fig. 4), and GII.3 VLP, access to M4 is restricted when VP1 is in the resting conformation. See lines 281-290 in the revised text.

- Check again carefully all the references! For instance, in reference 6, Mortality is certainly not the first author.

Thank you very much. We have now rectified it, ref 6, lines 446-449

Reviewer #3 (Remarks to the Author):

The manuscript by Salmen et al entitled 'A single nanobody neutralizes multiple epochally evolving human noroviruses by modulating capsid plasticity' presents X-ray structure of human norovirus GII.4 Sydney P-domain (part of the viral protein 1, VP1) in complex with a neutralizing llama-derived nanobody M4. The structure reveals a conserved epitope among GII.4 HuNoVs targeted by M4. Comparison of these results with the structure of the capsid (of viral like particles, VLPs) indicates that M4 binds to the raised VP1 conformation (GII.4 VP1 is dynamic and can adopt 'resting' and 'raised' conformations in the capsid). This is an important information as up to now most of the GII HuNoV neutralizing antibodies have been shown to block receptor binding. By additional assays authors have shown that M4 binding to the capsids leads to capsid dissociation. Nanobodies like M4 potentially could be further developed as antivirals against GII.4 noroviruses.

Major points

1. Authors should check they X-ray derived atomic model. Zero Ramachandran outliers do not fit well with the presence of other outliers (like RSRZ outliers) and the reported resolution 2.87 Å; in addition, reported Rfree is relatively high meaning that there is some discrepancy between the model and the experimental data. Authors should try to reduce Rfree even at the expense of few Ramachandran outliers. Zero Ramachandran outliers can be misleading (<https://doi.org/10.1016/j.str.2020.08.005>).

Thank you for this comment. We checked and reprocessed the data and noted that the Rama-Z score was indicative of a good-quality model falling within the acceptable range of +/- 2. (Rama-Z = -0.57 for the whole structure with N=1623). Table 1 is revised based on this reprocessing.

2. Description of endocytosis assay is insufficient, authors should revise.

We have updated the endocytosis assay with a more detailed method. Lines 404-412.

Specific comments

The title

Authors use a term 'multiple epochally evolving human noroviruses' in the title. It would be useful to explain for the reader what does in mean, for example in the introduction.

We have clarified this in the introduction. Lines: 51-53

Lines 117-120, 'A superposition of the structures of GII.4 Sydney P-domain alone and in complex with M4 and in complex with H-type HBGA showed that M4 binding does not result in any significant conformational changes in the P-domain, with a C α RMSD of 0.68 Å, and HBGA binding site is not affected (Fig. 2b).' Not clear for which structure comparison RMSD is shown, P-domain alone vs P-domain in complex with M4 or P-domain alone vs P-domain in complex with HBGA.

Thank you for your comment. We have clarified the sentence to specify that the root mean square deviation is between the matching Calpha atoms. See lines 121-122.

Fig 2 Grey is hard to distinguish from blue.

Thank you for the suggestion. As suggested, we have changed the structure color from gray to orange. See Fig 2b

Line 73: It would be good to know more on histo-blood group antigens (HBGAs), at least from the structural point of view.

Line 125: It is not clear what are P1 and P2 subdomains. Some schematics could help.

In response to these suggestions, we have added a new Supplementary Fig. 1 providing a schematic of residues that constitute different domains and HBGA binding site and appropriately referenced them in lines 70, 78, and 128 in the revised manuscript.

Line 139: 'shows that the identified M4 binding site in our crystallographic structure is highly conserved (72% to 94% similarity)' How the amino acid similarity is defined? If it is identity, then word 'identity' should be used.

We have revised the text to indicate that it is identity. See lines 142-143

Figure 4 a and b, the M4 binding site is not seen in these figures. In addition to showing atomic

structures inside the cryoEM density, authors should include visual for atomic models of P with/without M4 in context of the virion (based on the VLP structures) with highlighted (clearly visible) binding site for the nanobody M4. Perhaps different angle/view could help here.

Thank you for the suggestion. We remade the figures to improve the clarity of the structures. See Fig.2

Lines 229-231: 'We performed similar endocytosis experiments with GII.4 VLPs in the presence of M4 and observed the endocytic uptake is significantly diminished, consistent with capsid disassembly by M4 as a mechanism of neutralization (Sup. Fig. S3).' This experiment is not described in the results.

Thank you for noticing this. We have now included words describing this in the results section and improved the method section on the endocytosis experiments. See lines 201-209 (Results), lines 404-412 (Methods)

The discussion section looks like a mixture of results and discussion. Having results as a separate section, all results should be described in the results section.

Thank you for the suggestion. We have moved the endocytic entry results from the discussion into the results. See lines 201-209.

Line 284: 'reverse transcriptase quantitative PCR (RTqPCR)' should be 'reverse transcriptase quantitative PCR (RT-qPCR)'.

Thank you, we revised this line. Line 313 in the revised manuscript

Line 337: please add final molar concentrations for both VLPs and M4 to indicate molar ratio of nanobody vs its target (VLPs).

Added – lines 365-368

Lines 362-369: Description of endocytosis assay is insufficient. Please clarify what is FM1-43FX (ThermoFisher Scientific). Line 365: 'Briefly, HIE monolayers were 10 µg/mL of FM1-43FX for 10 min at 37 °C with either VLP alone or VLP pre-incubated with M4 for 1 h at 37 °C.' From this sentence it is not clear what has been done with HIE monolayers. It is not clear how authors measured fluorescence before quantitation.

Thank you for this comment, we have revised the methods with more details on quantitation. Lines 403-412.

The quality of the structure (PDB report): the atomic model shows zero (0) Ramachandran outliers at the resolution of 2.87 Å (somewhat low resolution for X-ray structures). The reported Rfree is somewhat high. Authors should try to improve their model by reducing Rfree even if it will introduce few Ramachandran outliers. Zero Ramachandran outliers can be misleading; authors should consult for example this paper <https://doi.org/10.1016/j.str.2020.08.005>.

Thank you. As per the suggestion, we reprocessed the data and updated the table. After reprocessing, the Rama-Z score is -0.57 for the whole structure (N=1623 atoms), which indicates the model is of good quality as this value is within the acceptable range of +/- 2. The Table 1 is now revised.

Reviewer comments, second round:

Reviewer #1 (Remarks to the Author):

I agree this is the first study to demonstrate a Nanobody neutralizes GII.4 in HIEs, but that is the main and only novelty.

It is not surprising at all that this Nanobody binds multiple GII.4 variants, since the binding region is highly conserved for all GII.4 sequences! It would be of novelty and importance if this Nanobody blocked and neutralizes numerous GII genotypes!

I am still not convinced that particle disassembly is the most accurate description of this Nanobody neutralization, because you are using VLPs not virions.

Particle disassembly for other antibodies and Nanobodies have already been published multiple times and the binding regions for these antibodies and Nanobodies all differ. This means, there are many regions on the capsid that disassemble when bound by these different antibodies and Nanobodies? It could also mean that immunity would often create some kind of particle disassembly?

Reviewer #2 (Remarks to the Author):

I consider that the authors have adequately responded to all questions.

Reviewer #3 (Remarks to the Author):

The authors have improved the manuscript 'A single nanobody neutralizes multiple epochally evolving human noroviruses by modulating capsid plasticity' according to reviewers comments. In my opinion, the manuscript in its current version can be published in the scientific journal.